

# Future tropospheric ozone budget and distribution over East Asia under a Net Zero scenario

Xuewei Hou[1], Oliver Wild[2], Bin Zhu[1], James Lee[3]

[1]Collaborative Innovation Center on Forecast and Evaluation of Meteorological Disasters, Key Laboratory of Meteorological Disaster, Ministry of Education (KLME), School of Atmospheric Physics, Nanjing University of Information Science and Technology, Nanjing, China

[2]Lancaster Environment Centre, Lancaster University, Lancaster, UK

[3]Department of Chemistry, University of York, York, UK

*Correspondence to*: Xuewei Hou (houxw@nuist.edu.cn)

**Abstract:** Under future net zero emission policies, reductions in emissions of ozone ($O_3$) precursors are expected to alter the temporal and spatial distribution of tropospheric $O_3$. In this study, we quantify changes in the tropospheric $O_3$ budget, spatiotemporal distribution of surface $O_3$ in East Asia and the contributions from regional emissions, intercontinental transport and climate change between the present day and 2060 under a net zero scenario, using the NCAR Community Earth System Model (CESM) with online tagging of $O_3$ and its precursors. The results reveal that the global tropospheric $O_3$ burden is likely to decrease by more than 20%, from 316 Tg in present day to 247 Tg in 2060, under a net zero scenario. The burden of stratospheric $O_3$ in the troposphere is expected to increase from 69 to 77 Tg. The mean lifetime of tropospheric $O_3$ increases by 2 days, ~10%. Changes in climate under a net zero pathway are relatively small, and only lead to small increases in tropospheric $O_3$. Over East China, surface $O_3$ increases in winter, due to the weakened titration of $O_3$ by NO associated with reduced anthropogenic NO emissions, and to enhanced stratospheric input. In summer, surface $O_3$ decreases by more than 30 ppbv, and peak concentrations shift from July to May. Local contributions from anthropogenic emissions to surface $O_3$ over East Asia are highest in summer, but drop substantially, from 30% to 14%, under a net zero scenario. The contribution of biogenic sources is enhanced, and forms the dominant contributor to future surface $O_3$, especially in summer, ~40%. This enhanced contribution is mainly due to the increased $O_3$ production efficiency under lower anthropogenic precursor emissions. Over Eastern China, local anthropogenic contributions decrease from 50% to 30%. The decreases in surface $O_3$ are strongly beneficial, and are more than sufficient to counteract the increases in surface $O_3$ observed in China over recent years. This study thus highlights the important co-benefits of net zero policies that target climate change in addressing surface $O_3$ pollution over East Asia.

**Keywords:** Tropospheric $O_3$; SSP1-1.9 pathway; net zero; $O_3$ budgets; stratospheric contribution

**1 Introduction**



Although ozone ($O_3$) occurs naturally in small quantities in the lower troposphere, unhealthy levels of
tropospheric $O_3$ are created when high levels of anthropogenic pollutants, such nitrogen oxides (NOx),
and volatile organic compounds (VOCs) are oxidized in the presence of solar radiation. This excess $O_3$
acts as a pollutant and greenhouse gas, contributing to harmful smog that damages human health and
ecosystems (Jerrett et al., 2009; Malley et al., 2017; Emberson, 2020) and contributes to higher
temperatures near Earth's surface (Myhre et al., 2013; Stevenson et al., 2013). The relatively short
lifetime of $O_3$ in the troposphere (~3 weeks, Young et al., 2013) means that it is classified as a Near Term
Climate Forcer (NTCF), having an important influence on climate over shorter timescales. Tropospheric
$O_3$ also is an oxidant and a precursor for the hydroxyl (OH) radical (Griffiths et al., 2021). OH (and by
implication $O_3$) controls the lifetime of methane (Voulgarakis et al., 2013), the second most important
anthropogenic greenhouse gas after carbon dioxide (Myhre et al., 2013). Oxidant levels mediate the
formation of secondary aerosols such as sulfate and nitrate and play a major role in the aerosol budget
and burden with important consequences for radiative forcing (Shindell et al., 2009; Karset et al., 2018).
Understanding how tropospheric $O_3$ changes is important for both future air quality and climate (Turnock
et al., 2019).
A multi-model assessment of future changes in tropospheric $O_3$ was made in the Atmospheric Chemistry
and Climate Model Intercomparison Project (ACCMIP), using future changes in climate and $O_3$
precursor emissions from the Representative Concentration Pathways (RCPs) (Lamarque et al., 2013).
The models participating in ACCMIP projected changes in global annual mean surface $O_3$ concentrations
between 2000 and 2030 of ±1.5 ppbv under the different RCPs (Young et al., 2013). More recent single
model estimates by O'Connor et al. (2014) and Kim et al. (2015) predict surface $O_3$ responses across the
different RCPs of between −4.0 and +2.0 ppbv by 2050 relative to 2000. The global annual mean
tropospheric $O_3$ burden was projected to change by between −18% and +20% from 2000 to 2100 under
the different RCPs (Cionni et al., 2011; Kawase et al., 2011; O'Connor et al., 2014; Young et al., 2013).
Whether tropospheric $O_3$ increases or decreases in future is dependent on the climate mitigation measures
that are taken. In preparation for the sixth Coupled Model Intercomparison Project (CMIP6), a new set
of future pathways were created. Five different socio-economic pathways (SSPs) were developed with
centennial trends based on different combinations of social, economic and environmental developments
(O'Neill et al., 2014). Different levels of emissions mitigation were included within each SSP to meet
particular climate and air pollution targets (Rao et al., 2017; Riahi et al., 2017). They incorporate stronger
links between socio-economic development patterns and climate change risks than previous assessments
and provide better hypothetical scenarios for future projections. The five most widely-used scenarios are
SSP1-1.9, SSP1-2.6, SSP2-4.5, SSP3-7.0, and SSP5-8.5, where SSP1-SSP5 represent differing
development storylines and the suffix 1.9~8.5 indicates the radiative forcing values ($W/m^2$) at the end of
the 21st century compared with those before the Industrial Revolution. These pathways provide a good
foundation for assessment of air quality, radiative forcing, ecological environmental effects and human
health effects in the future. Many studies have focused on the pessimistic SSP3-7.0 scenario reflecting
regional rivalry, and Griffiths et al. (2021) found that the tropospheric $O_3$ burden increases from
$356 \pm 31$ Tg in present day to $416 \pm 35$ Tg in 2100 under this pathway. The sustainability-focused SSP1-
1.9 pathway is the scenario mostly closely aligned with recent pledges aiming at net zero greenhouse gas



emissions, limiting warming to 1.5°C by 2100, but the impacts of this pathway on tropospheric $O_3$ are
less well studied and remain unclear.
In East Asia, surface $O_3$ has increased rapidly since 2000 (Lu et al., 2020), and is expected to increase
by another ~10 ppbv by 2050 (Wang et al., 2013; Zhu and Liao, 2016; Hong et al., 2019). In September
2020, China committed to achieve carbon neutrality by 2060, following the commitments of many
developed countries to achieve net zero emissions by 2050. The effect of these strong mitigation
measures on surface $O_3$ has not been explored thoroughly, but the proposed emission pathway to net zero
loosely aligns with the SSP1-1.9 pathway. Turnock et al (2019) showed large reductions of >8 ppbv in
surface $O_3$ over East Asia by 2050 along this pathway due to large reductions in precursor emissions and
$CH_4$. The study also shows that any benefits to surface $O_3$ from reducing local emission sources over
East Asia could be offset by intercontinental transport of $O_3$ formed from sources remote to the region
and from global $CH_4$ sources. This analysis used an $O_3$ parameterization to rapidly assess changes in $O_3$
and source attribution (Wild et al., 2012; Turnock et al., 2018), but did not account for changes in climate,
stratosphere-to-troposphere exchange, or chemical regime. Wang and Liao (2022) found that the annual
mean contribution of Southeast Asia to surface MDA8 $O_3$ in China is 3-19 μg m$^{-3}$, about 2-10 ppbv, and
this contribution is reduced in future along the SSP1-1.9 pathway. However, this study used fixed
meteorological parameters for 2015, preventing quantification of regional transport contributions and
stratosphere-tropospheric exchange. Other recent assessments exploring the implications of carbon
neutrality in China have suggested that there may be reductions in MDA8 $O_3$ of more than 30 μg m$^{-3}$ (15
ppb) by 2060 and that regional mean $O_3$ concentrations may decline to 63.0 μg m$^{-3}$ (Shi et al., 2021;
Wang and Liao, 2022; Xu et al., 2022).
While previous studies have quantified possible changes in surface $O_3$ under carbon neutrality, the wider
impact on the global tropospheric $O_3$ budget and the changing contributions of different sources remain
unclear. In this study, we quantify the changes in surface $O_3$ over East Asia, and especially over Eastern
China which currently has high anthropogenic emissions, and the contribution of different sources based
on emissions and climate change along the SSP1-1.9 pathway, using the NCAR Community Earth
System Model (CESM) with online tagging of $O_3$ and its precursors. We present a self-consistent
assessment of the changes in surface $O_3$ associated with changes in emissions and climate, along with
the first attribution of these changes. The paper is organized as follows. Section 2 describes the model
configurations, experimental settings, $O_3$ tagging method, and evaluation datasets. In section 3, $O_3$ and
$NO_x$ in present day simulations is evaluated against observations. In section 4, changes in tropospheric
$O_3$ under the net zero scenario are presented. In section 5, the contribution of $O_3$ chemistry and
intercontinental transport are discussed under present day and future conditions. We close with a
summary in section 6.
**2 Materials and methods**
**2.1 Model configurations and experiments**





The NCAR Community Earth System Model (CESM) is a coupled climate model incorporating
components for simulating the Earth's atmosphere, ocean, land, land-ice, and sea-ice (e.g., Neale et al.,
2013; Lamarque et al., 2012; Tilmes et al., 2015; Danabasoglu et al., 2020), allowing fundamental
research into the Earth's past, present, and future climate states. The experiments here use CESM version
1.2.2        (https://www.cesm.ucar.edu/models/cesm1.2/)         and       the       latest       version       2.2.0
(https://www.cesm.ucar.edu/models/cesm2/), to reproduce present-day $O_3$ mixing ratios and to predict
$O_3$ responses to emissions and climate in the future along the SSP1-1.9 pathway. All model simulations
are performed with prescribed sea surface temperatures and sea ice distribution data for climatological
conditions in present day and future net zero, since we focus on the atmospheric component. Dry
deposition of gases and aerosols are implemented in the Community Land Model (Oleson, 2010) as
described in Lamarque et al. (2012).
Atmospheric chemistry of gas phase and aerosol species in the global Community Atmosphere Model
(CAM version 4, Neale et al., 2013; CAM version 6, Danabasoglu et al., 2020), the atmospheric
component of the Community Earth System Model (CESM), is represented by CAM-chem. CAM-chem
provides the flexibility of using the same code to perform climate simulations (online) and simulations
with specified meteorological fields (offline). The chemical mechanism is based on the Model for Ozone
and Related chemical Tracers (MOZART), version 4 mechanism for the troposphere (Emmons et al.,
2010), extended for stratospheric chemistry (Kinnison et al., 2007), with further updates as described in
Lamarque et al. (2012), including additional reaction rate updates following JPL-2010 recommendations
(Sander et al., 2011).
In this paper, offline simulations are used to investigate the effect of emission changes on tropospheric
$O_3$ under fixed meteorological parameters, while online simulations are used for the effects of emission
and climate changes with two-way feedback of atmospheric components and meteorological parameters.
Two different versions of CESM are used in this study due to the application of online tagging of $O_3$ and
its precursors, which is only fully tested and evaluated in CESM1. All simulations discussed in this paper
are performed at a horizontal resolution of 1.9° (latitude) and 2.5° (longitude). The model has 26 vertical
levels in the online configuration and 56 levels in the offline configuration using specified meteorological
fields; in all these cases, the model extends to approximately 4 hPa (≈40 km). Offline simulations were
driven by Modern Era Retrospective analysis for Research and Applications (MERRA2) meteorology
(Rienecker et al., 2011). Simulations using present-day emissions (2015) are labelled PD, while those
using future net zero emissions (2060) are labelled NZ, and these are prefixed with online or offline
depending whether the model is run online or driven by MERRA2 meteorology. To ensure the stability
of the response to climate change, the future online simulations are run for 15 years, with the first ten
years as spin-up. The $CH_4$ concentrations are prescribed following the SSP1-1.9 pathway using a fixed
lower boundary condition. A summary of the simulations is provided in Table 1.

144                              Table 1 Experimental settings

| Case-name | Climate Change and Emissions | Emissions |
|---|---|---|



| | online-PD | online-NZ | offline-PD | offline-NZ |
|---|---|---|---|---|
| Model | CESM1.2.2 | CESM1.2.2 | CESM2.2.0 | CESM2.2.0 |
| Component | FMOZ | FMOZ | FCSD | FCSD |
| Physics | CAM4 | CAM4 | CAM6 | CAM6 |
| Chemical mechanism | tropospheric chemistry with bulk aerosols, MOZART-4 | | troposphere/stratosphere chemistry with simplified VBS-SOA, MOZART-TS1 | |
| Dynamics | Free-running | Free-running | Merra2 Nudging | Merra2 Nudging |
| Spin-up | 2012-2014 | 2050-2059 | 2014 | 2014 |
| Analyzed Year | 2015-2016 | 2060-2064 | 2015 | 2015 |
| Resolution | 1.9°×2.5° with 26 levels | 1.9°×2.5° with 26 levels | 1.9°×2.5° with 56 levels | 1.9°×2.5° with 56 levels |
| Emission — in China | 2015- DPEC | 2060-DPEC | 2015-CMIP6 | 2060-DPEC |
| Emission — Outside China | 2015-SSP119 | 2060-SSP119 | 2015-CMIP6 | 2060-SSP119 |
| CH$_4$ | 2015-SSP119 | 2060-SSP119 | 2015-SSP119 | 2060-SSP119 |
| Tagging O$_3$ sources | TOAST | | O3S | |

## 2.2 Emissions

For this analysis, we use estimates of global future anthropogenic and biomass burning emissions and future abundances of greenhouse gases and aerosols provided by the SSP1-1.9 pathway (https://esgf-node.llnl.gov/projects/input4mips/) along with more recent estimates for China using the Ambitious-pollution-Neutral-goals scenario from the Dynamic Projection model for Emissions in China (DPEC, http://meicmodel.org/). The SSP1-1.9 pathway results in a climate radiative forcing of 1.9 W m$^{-2}$ by 2100 under the sustainable development path. The SSP1-1.9 pathway is a strong pollution control scenario and is the only route to limit the global average temperature increase since the preindustrial period to 1.5°C by 2100 (O'Neill et al., 2014; Rao et al., 2017; Riahi et al., 2017). The emissions inventory includes monthly O$_3$ precursors, aerosols, and their precursors (NO$_x$, CO, non-methane volatile organic (VOCs), sulfur dioxide (SO$_2$), ammonia (NH$_3$), black carbon (BC), organic carbon (OC), dimethyl sulphide (DMS)), and concentrations of greenhouse gases, such as CH$_4$. Biogenic emissions of VOCs are calculated online in CESM using the Model of Emissions of Gases and Aerosols from Nature model (MEGAN; Guenther et al., 2006; 2012). We use emissions for the years 2015 and 2060. Over China, the anthropogenic emissions are replaced by the Ambitious-pollution-Neutral- goals scenario from DPEC (Tong et al., 2020; Cheng et al., 2021). This considers a scenario in which China achieves carbon neutrality by 2060. The combined emissions distribution for NO$_x$ and its changes in future are shown in Figure S1. The total annual mean surface emissions of key pollutants from anthropogenic (ANT),





biomass burning (BB) and biogenic (BIO) sources for the present day (2015) and future net zero (2060)
over the globe and in East Asia are listed in Table 2.
The global anthropogenic emissions of all $O_3$ precursors are significantly reduced in the net zero scenario.
Due to strict control policies on pollutants emissions and changes in technology and behavior, global
anthropogenic NO emissions decrease from 87 Tg yr$^{-1}$ in present day to 19 Tg yr$^{-1}$ in 2060, and total
anthropogenic VOCs emissions decrease from 125 Tg yr$^{-1}$ to 28 Tg yr$^{-1}$. Biomass burning emissions
decrease slightly. Natural NO soil emission, VOCs biogenic emission, and CO ocean emission are
assumed not to change in this study as changes in land use are relatively small. Anthropogenic emissions
over East Asia account for more than 35% of the global total, with biomass combustion emissions
accounting for a smaller proportion, ~10%, and natural emissions of NO, VOCs and CO accounting for
~20%. The decrease of anthropogenic emissions over East Asia (about 80% for NO) is greater than the
global average, >70%, which may due to the high present day emissions over the region, especially in
Eastern China. The global $CH_4$ concentration decreases from the current 1831 ppbv to 1312 ppbv, due
to the lower global $CH_4$ emissions under net zero.
Table 2 Annual mean time-varying surface emissions of NOx, VOCs, CO, sulfur dioxide ($SO_2$), black
carbon (BC), and organic carbon (OC) from anthropogenic (ANT), biomass burning (BB) and biogenic
(BIO) emissions for the present day (2015) and future (2060, net zero) in East Asia and over globe.
Annual mean surface $CH_4$ mixing ratios (ppbv) are also shown.

| Emission (Tg yr$^{-1}$) | | Globe | | East Asia | |
|---|---|---|---|---|---|
| | | Present Day | Net Zero | Present Day | Net Zero |
| NO | ANT | 87.5 | 19.1 | 36.9 | 7.5 |
| | BB | 8.9 | 7.5 | 0.7 | 0.5 |
| | Soil | 10.6 | 10.6 | 2.3 | 2.3 |
| | Total | 106.9 | 37.2 | 39.8 | 10.2 |
| VOCs | ANT | 125.0 | 27.5 | 42.9 | 11.0 |
| | BB | 66.6 | 50.2 | 6.3 | 4.0 |
| | BIO | 868.5 | 868.5 | 111.0 | 111.0 |
| | Total | 1060.1 | 946.2 | 160.3 | 126.0 |
| CO | ANT | 559.8 | 151.7 | 266.9 | 72.7 |
| | BB | 325.5 | 248.2 | 30.2 | 18.9 |
| | Ocean | 20.0 | 20.0 | 1.3 | 1.3 |
| | Total | 905.2 | 419.9 | 298.4 | 92.9 |
| $SO_2$ | ANT | 105.2 | 18.8 | 40.2 | 4.8 |
| | BB | 2.1 | 1.7 | 0.2 | 0.1 |



| | | | | | |
|---|---|---|---|---|---|
| | Total | 107.4 | 20.5 | 40.4 | 5.0 |
| BC | ANT | 6.7 | 1.0 | 3.0 | 0.3 |
| | BB | 1.7 | 1.4 | 0.2 | 0.1 |
| | Total | 8.4 | 2.3 | 3.1 | 0.4 |
| OC | ANT | 16.5 | 4.1 | 6.5 | 1.4 |
| | BB | 15.1 | 10.6 | 1.6 | 0.9 |
| | Total | 31.5 | 14.8 | 8.1 | 2.3 |
| $CH_4$ (ppbv) | | 1830.5 | 1312.2 | 1860.8 | 1337.3 |


**2.3 Tagging of ozone**
In this study, we used the Tropospheric Ozone Attribution of Sources with Tagging (TOAST) ozone
methodology in CESM1.2.2 previously described by Butler et al. (2018, 2020) to perform separate source
attributions of ground-level $O_3$ to $NO_x$. The parameterizations based on the work of Butler et al. (2018,
2020) include tagging biogenic, biomass burning and anthropogenic emissions of NOx or VOCs by their
geographical source regions. This tagging methodology allows us to examine the seasonal cycle of the
surface $O_3$ attribution in receptor regions using those defined in the Hemispheric Transport of Air
Pollutants Phase 2 (HTAP2, Janssens-Maenhout et al., 2015; Koffi et al., 2016). We considered 16
sources, including 11 geographical source regions for anthropogenic NOx, shown in Table 3 and Figure
1, NOx emissions from biogenic sources (BIO), biomass burning (BB), aircraft (AIR) and lightning (LIG),
and $O_3$ originating in the stratosphere (STR).
Table 3 Source sector tagging of anthropogenic NOx emissions by geographical source region, NOx
emissions from biogenic burning, soil emission, aircraft, and lightning, and the contribution of
stratospheric $O_3$ input.

| ID | Geographical region, NOx | ID | Geographical region, NOx | ID | Source |
|---|---|---|---|---|---|
| OCN | Oceans | NAF | Northern Africa | BIO | Biogenic NOx |
| NAM | N. America | MDE | Middle East | BB | Bioburn NOx |
| EUR | Europe | CAS | Central Asia | AIR | Aircraft NOx |
| SAS | South Asia | SEA | South East Asia | LIG | Lightning NOx |
| EAS | East Asia | RBU | Russia, Belarus, Ukraine | STR | Stratospheric O3 |
| RST | Rest of World | | | | |




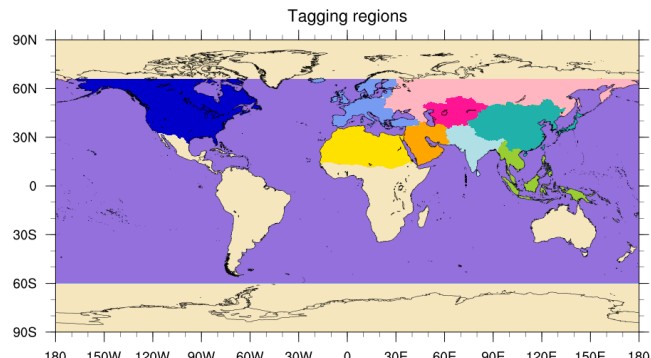

Figure 1 Geographical source regions for tagging anthropogenic NO$_x$ emissions in this study as defined in HTAP Phase 2.

**2.4 Measurement Data**

To evaluate tropospheric column O$_3$ in the model simulations, we used a present-day satellite dataset of tropospheric column O$_3$, which was derived by combining retrievals from the Aura Ozone Monitoring Instrument (OMI) and Microwave Limb Sounder (MLS) observations (https://acd-ext.gsfc.nasa.gov/Data_services/cloud_slice/). More details about the generation of this dataset are provided by Ziemke et al., (2011). The dataset resolution used in this study is 1° (Latitude) ×1.25° (Longitude) and the year is 2015. The monthly-mean thermal tropopause pressure was used to separate tropospheric and stratospheric O$_3$ for the model results and satellite observations.

A High-resolution Air Quality Reanalysis Dataset over China (CAQRA, Kong, et al., 2020; Tang et al., 2020 a, b) was used to evaluate the simulated present day surface O$_3$ over China (https://doi.org/10.11922/sciencedb.00053). This dataset was generated by assimilating surface observations from the China National Environmental Monitoring Centre (CNEMC) into the Nested Air Quality Prediction Modeling System (Tang et al., 2011, Wang et al., 2000), and it provides self-consistent concentration fields of O$_3$ in China from 2013 to 2019 at high spatial (15 km) and temporal (1 h) resolutions. The year used in this study is 2015.

In addition, monthly observational surface O$_3$ concentration were taken from 12 regional stations of the Acid Deposition Monitoring Network in East Asia (EANET; https://www.eanet.asia/) for 2015: Rishiri, Ochiishi, Tappi, Sado-Seki, Happo, Oki, Yusuhara, Hedo, Mondy, Listvyanka, Kanghawa, and Chenju. The locations and altitudes of these sites are shown in Figure S2.

**3. Tropospheric ozone evaluation**

We compared the simulated monthly mean tropospheric column O$_3$ (TCO) with that derived from OMI/MLS for January and July in 2015 (Figure 2). The model captures the general features of the observed tropospheric column, reproducing the seasonal pattern, with a minimum of 15 DU at 180°E in the tropics during January and a maximum of >50 DU in northern hemisphere mid-latitudes during July.



The highest values in the northern mid-latitudes are overestimated in both offline and online simulations,
especially during July. In the simulations, TCO was calculated by integrating the $O_3$ from the surface to
the tropopause. Some of the differences of the simulated TCO with OMI/MLS may be due to the
relatively coarse vertical resolution of the model, which is averages 183 hPa for the online model near
the tropopause (194 hPa in NCEP reanalysis datasets). Uncertainty in the satellite dataset (exceeding
5DU in high latitudes, Ziemke et al. 2011) might also contribute to these differences. The global
(60°S~60°N) annual mean tropospheric $O_3$ columns from the offline and online simulations are 31.3 and
32.2 DU, respectively, which match those from OMI/MLS (31.6 DU) and the ACCMIP models mean
value (30.8 DU, Young et al., 2013) well. The online simulated tropospheric ozone column on global
annual average is the highest, due to the coarser vertical resolution in online simulation (Lamarque et al.,

235  2012).

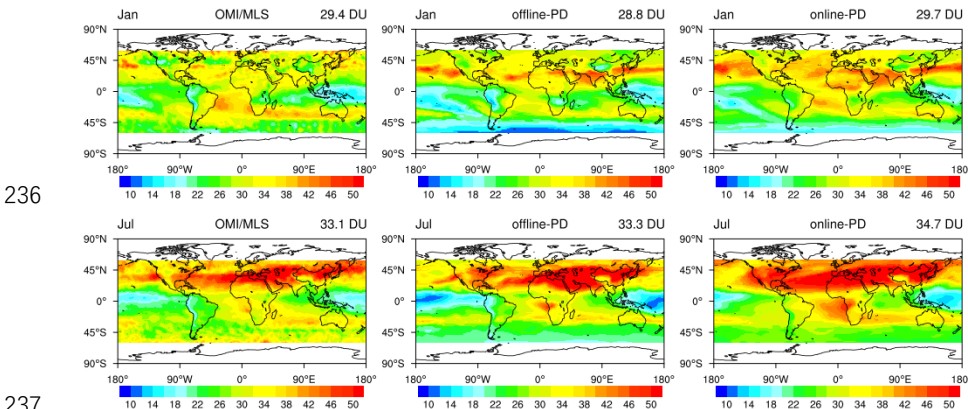



Figure 2 Tropospheric column $O_3$ (DU) from OMI/MLS (left), offline (middle) and online (right)
simulations for January and July under present day conditions.

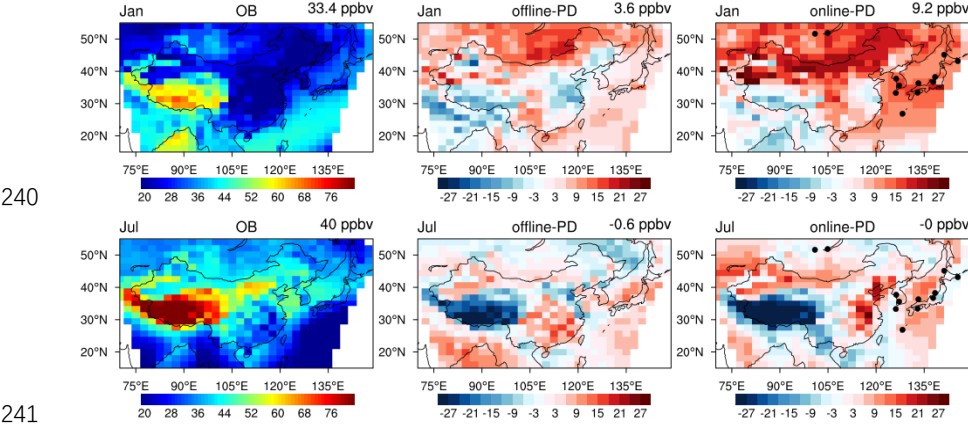



Figure 3 Surface $O_3$ mixing ratios in East Asia (ppbv) from the CAQRA reanalysis (left) and the biases
from offline (middle) and online (right) present day simulations in January and July. The biases are





244 simulations minus observations, and black dots show the locations of EANET observation sites. The

245 values in the right corner of each sub-figure are the regional mean for East Asia (15~55°N, 70~149°E).

246 As shown in Figure 3, surface $O_3$ shows substantial seasonal variations with low concentrations in winter

247 and high concentrations in summer. The spatial distributions of simulated surface $O_3$ concentrations

248 match the observations well. The online simulated surface $O_3$ (ppbv) is overestimated by 9.2 ppbv on

249 average in winter, especially in Mongolia, north and middle of China, Korea, and Japan, while the offline

250 simulation is much closer to the observation with a bias of 3.6 ppbv. The comparison of simulated surface

251 $O_3$ (ppbv) with EANET observations show that the simulations reproduce the seasonal variations at these

252 12 sites (Figure S2 in the Supplementary Material). In general, the performance of these simulations is

253 very similar to those from other chemical model studies (Li et al., 2019; Young et al., 2018).

**4 Tropospheric ozone budgets and distributions under the Net Zero scenario**

255 An overview of the global model diagnostics for the simulation experiments is given in Table 4.

256 Tropospheric $O_3$ burden and budget terms for present-day conditions in this study match previous results

257 well. Under net zero, the chemical production decreases from 5038 to 3392 Tg($O_3$) yr$^{-1}$, and the chemical

258 loss decreases from 4641 to 3311 Tg($O_3$) yr$^{-1}$. The net chemical tendency of tropospheric $O_3$ (NetChem

259 in Table 4) drops substantially, decreasing from the current 397 Tg($O_3$) yr$^{-1}$ to 81 Tg($O_3$) yr$^{-1}$, due to the

260 large reduction in $O_3$ precursor emissions (Table 2). This results in an increase in the lifetime of

261 tropospheric $O_3$ from 20 days to 22 days. The residual term, which principally reflects net transport from

262 the stratosphere, increases from the current 595 to 626 Tg($O_3$) yr$^{-1}$. The global tropospheric $O_3$ burden

263 decreases by about 20%, from 316 Tg to 247 Tg, bringing it close to the mean burden of 239±22 Tg

264 estimated for the pre-industrial period (Young et al., 2013). The burden of $O_3$ of stratospheric origin in

265 the troposphere ($O_3S$) increases from 69 Tg to 77 Tg. This increased stratospheric contribution may be

266 due to the enhancement of stratospheric circulation and increased stratosphere-troposphere exchange

267 caused by climate change (Sudo et al., 2003; Lu et al., 2019). In addition, the longer chemical lifetime

268 allows stratospheric $O_3$ to persist for longer in the troposphere, enhancing the stratospheric contribution.

269 Over East Asia, the net photochemical production of tropospheric $O_3$ also decreases significantly, from

270 the current 227 Tg($O_3$) yr$^{-1}$ to 137 Tg($O_3$) yr$^{-1}$ under net zero, but the reduction is less than the global

271 average. The negative "Residual" budget term for East Asia indicates that East Asia is a net outflow

272 region for tropospheric $O_3$, and this outflow is weakened in the future, from 89 Tg($O_3$) yr$^{-1}$ under present

273 day conditions to 38 Tg($O_3$) yr$^{-1}$ under net zero, reflecting the reduced regional production. The

274 tropospheric $O_3$ burden in East Asia decreases from 25 Tg to 19 Tg, while the burden of $O_3$ from the

275 stratosphere increases slightly from 5 Tg to 6 Tg. The tropospheric $O_3$ lifetime in East Asia is 15 days,

276 slightly lower than the global average due to the faster photochemical processing under relatively high

277 anthropogenic emissions. But the increase of ~2 days matches that of the global average.



Table 4 Global tropospheric $O_3$ burden (Tg) and budget terms (Tg yr$^{-1}$) in chemical transport models.

| Models | Prod | Loss | NetChem | Residual | DryDep | Burden ($O_3/O_3S$) | Lifetime (days) | Reference |
|---|---|---|---|---|---|---|---|---|
| **Globe** | | | | **STE** | | | | |
| 33 | 3948±761 | 3745±554 | 245±346 | 636±273 | 902±255 | 307±38 | 21-25 | Wild (2007) |
| 17 | 4465±514 | 4114±409 | 396±247 | 529±105 | 949±222 | 314±33 | 22±2 | Stevenson et al. (2006) |
| 15 | 5110±606 | 4668±727 | 442±309 | 552±168 | 1003±200 | 344±39 | 22±2 | Young et al. (2013) |
| **PD** | **5038** | **4641** | **397** | **595** | **992** | **316/69** | **20** | **This study** |
| **NZ** | **3392** | **3311** | **81** | **626** | **707** | **247/77** | **22** | **This study** |
| **East Asia** | | | | **Transport** | | | | |
| **PD** | **682** | **455** | **227** | **-89** | **138** | **25/5** | **15** | **This study** |
| **NZ** | **430** | **293** | **137** | **-38** | **99** | **19/6** | **17** | **This study** |

Prod for chemical production, Loss for chemical loss, Prod-Loss for net chemical production (NetChem)
and DryDep for dry deposition; Residual is the term balance by Residual=Loss-Prod+DryDep. Units of
Prod, Loss, NetChem, Residual and DryDep are in Tg($O_3$) yr$^{-1}$, Burden in Tg($O_3$), and Lifetime in days.
The climatological pressure tropopause is used. PD is the online present day experiment simulation. NZ
is online net zero experiment simulation.

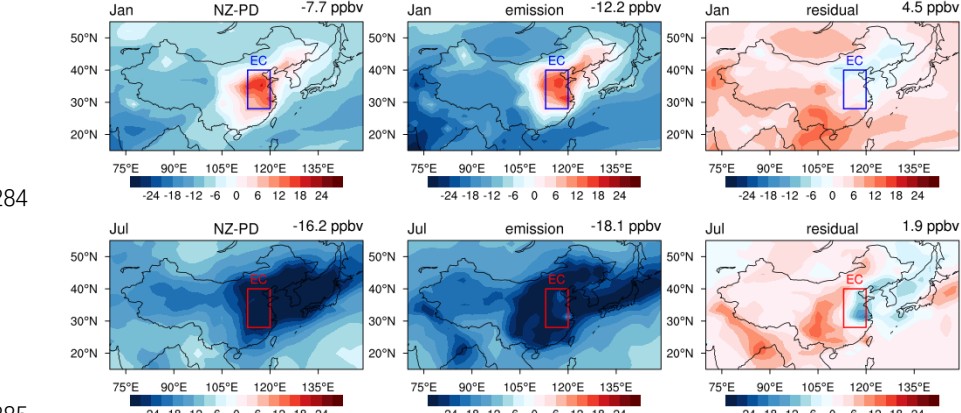

Figure 4 Changes in surface O$_3$ mixing ratio (ppbv) over China in January and July between present day and net zero (online-NZ minus online-PD, left), and changes due to emissions (offline-NZ minus offline-PD, middle) and the residual (left minus middle panel, right). The values in the right corner of each panel are the regions mean over East Asia (15°~55°N, 70°~149°E). The frame is the region of Eastern China (EC, 28°~40°N, 113°~120°E).

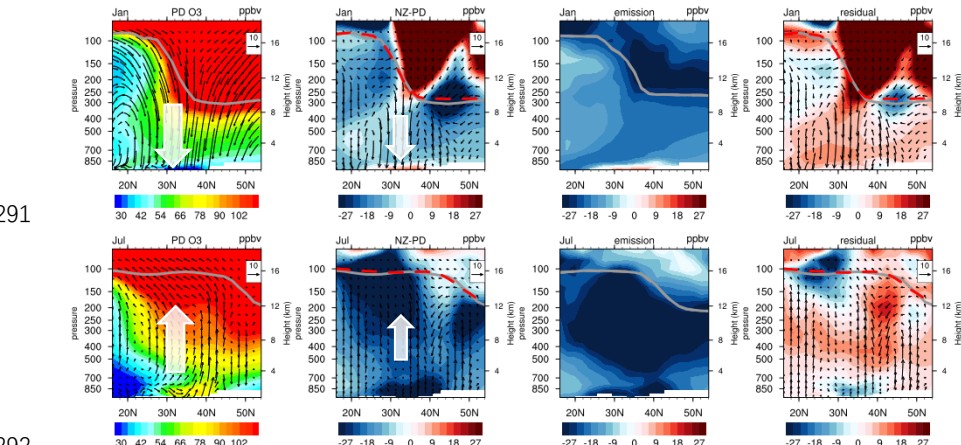

Figure 5 Zonal mean O$_3$ cross section (ppbv) and wind speed (vectors, v:m s$^{-1}$, w:*(-500) pa s$^{-1}$) over Eastern China (longitudes 111-122°E) in January and July under present day (online-PD, left), the changes in O$_3$ and wind speed (second panels) and changes due to emissions (third panels) and the residual (second panels minus third panels, right). Grey lines show the tropopause location under present day conditions; the red dashed lines show the tropopause location under net zero.

The changes in surface O$_3$ over East Asia between 2015 and 2060 in winter and summer are shown in Figure 4. The left panels show the changes in surface O$_3$ under net zero (online-NZ minus online-PD), which include the effects of climate change and emissions changes. The middle panels show the changes in surface O$_3$ under the effect of emissions changes only (offline-NZ minus offline-PD). The right panels



show the residual changes in surface $O_3$ which reflect the effect of climate change, but are also influenced
by differences in model setup between the online and offline simulations (left panels minus middle
panels). Surface $O_3$ decreases in East Asia under net zero, with a mean reduction of 7.7 ppbv in winter
and a greater reduction of 16.2 ppbv in summer. Turnock et al. (2019) estimated an annual mean
reduction of 8 ppbv in 2050 along the SSP1-1.9 pathway, slightly less than we find here. However, we
have used the more stringent DPEC Ambitious-pollution-Neutral-goals emission scenario for China
rather than the standard SSP1-1.9 pathway and we note that anthropogenic NO emissions in China are 2
Tg (NO) yr$^{-1}$ lower in this scenario than those in SSP1-1.9. Surface $O_3$ over Eastern China and South
Korea increases in winter in these scenarios, driven by the reduction in emissions (left and middle panels).
This increase in surface $O_3$ is caused by a weakening of titration under lower regional NO emissions. The
influence of climate change on surface $O_3$ is relatively weak, and leads to an increase of surface $O_3$ in
most parts of East Asia (right panels). This is partly due to enhanced vertical circulation leading to an
increased contribution from stratospheric $O_3$ (Akritidis et al., 2019; shown in Figure S3). Xu et al. (2022)
also showed that emission reduction is far more effective than climate change in improving air quality
($PM_{2.5}$ and $O_3$) over East Asia under a carbon neutral reduction pathway. Here we will use tagging
simulations to quantify the contributions of different sources to surface $O_3$ changes over East Asia,
especially over Eastern China where surface $O_3$ increases in winter and decreases in summer.
It can be seen from the vertical distribution of $O_3$ and circulation (shown in the first panels of Figure 5)
that the $O_3$ concentration increases with altitude under present day conditions. At the same altitude, the
$O_3$ concentration is higher in middle and high latitudes than in low latitude. In winter, there is strong net
descent of air over eastern China (30~40°N), which weakens in spring, and turns to updraft in summer.
These may be due to the weakened Brewer-Dobson circulation and strengthened convection (Butchart,
2014; Wild and Akimoto, 2001). As shown in the second panels of Figure 5, there is a net decrease in
tropospheric $O_3$ in future, with an increase only seen near 30°N very close to the surface. In summer, the
reduction in tropospheric $O_3$ is greatest, especially near the tropopause where it exceeds 30 ppbv. In
addition, due to the temperature increase and circulation enhancement in the future, the tropopause height
increases, especially in the mid-latitude region in winter where the increase is about 7 hPa. As seen from
the third panels of Figure 5, the reduction of emissions from aircraft (NO emissions in Figure S1) leads
to a reduction in $O_3$ production, and the $O_3$ concentration near the tropopause decreases substantially in
the future. However, other factors such as climate change (the fourth panels in Figure 5) lead to increases
in tropospheric $O_3$ by 2060.

**5 The contribution of $O_3$ chemistry and intercontinental transport**

Surface $O_3$ shows substantial seasonal variation over East Asia with a peak in spring, as shown in Figure
6a. It reaches a maximum (56 ppbv) in March and is lowest (41 ppbv) in August under present day
conditions. Under net zero, the concentration of surface $O_3$ is lower throughout the year, and while the
peak is still in March, the mixing ratio drops to 43 ppbv. The decrease is greatest in July, 16 ppbv, which
reflects weaker chemical production in summertime under lower future emissions (Figure 6c). In contrast,
surface $O_3$ over Eastern China is highest (71 ppbv) in July and lowest (21 ppbv) in December under
present day conditions (Figure 6b). Under net zero, surface $O_3$ increases in winter and decreases in
summer, and the peak shifts from July to May, due to the changes in $O_3$ precursors emissions (Bowman
et al., 2021). The decrease is highest in July, as seen over the wider East Asian region, but is twice as



large, at 34 ppbv, reflecting the stronger present-day emissions over Eastern China. There is a substantial
increase in $O_3$ in January of 12 ppbv, reflecting reduced titration by NO. The concentration of surface
$NO_x$ decreases more than 60%, and by an even larger factor in winter (~90%, 14 ppbv); and its seasonal
variation is reduced which accounts for the reduction in anthropogenic emissions (Figure 6d). In terms
of the $O_3$ chemical budget, local chemical production and destruction are both reduced in the future. The
peak in net $O_3$ chemical production still occurs in summer which highlights that photochemical processes
continue to dominate the seasonal variation of surface $O_3$ in Eastern China in future. However, the net
chemical destruction that currently occurs in winter is replaced with a small net $O_3$ production (Figure
6d), reflecting the reduced titration of $O_3$ by NO under future emissions, which are very greatly reduced
under net zero.
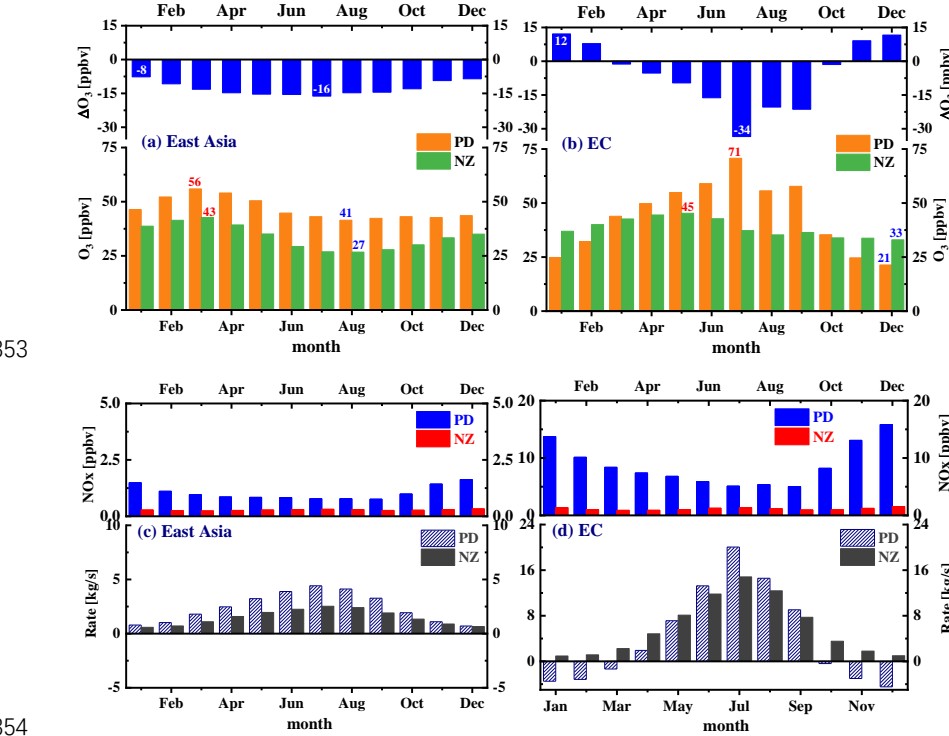

Figure 6 Comparison of $O_3$ (a, b), $NO_x$ and net $O_3$ chemical tendency (c, d) at the surface under present
day and net zero conditions over East Asia (left) and Eastern China (right). Results are from the online

simulations (online-PD and online-NZ). Maximum and minimum $O_3$ mixing ratios are highlighted in

red and blue, respectively, and the largest and smallest $O_3$ changes are indicated in white.

We quantify the contributions of regional transport and stratospheric input to surface $O_3$ on a monthly
basis in Figure 7. We find that the contribution of anthropogenic NO emissions from East Asia (EAS) is
highest, especially in summer when it reaches 30%. The contribution from biogenic NO emissions (BIO)
is also important, exceeding 10% in summer. The contributions from the ocean (OCN) and from



363 anthropogenic NO emissions over South Asia (SAS) show little seasonal variation, both contributing 10-

364 15%. Under net zero (Figure 7c), the contribution from East Asia drops dramatically, to 14% in summer,

365 due to the reduced emissions of $O_3$ precursors. The contribution of biogenic sources is enhanced, and

366 forms the dominant contributor to surface $O_3$ under net zero, especially in summer, ~40%. The emissions

367 from biogenic sources are changed slightly in this study. The enhanced contribution of biogenic sources

368 is mainly due to the increased $O_3$ production efficiency, which is a consequence of lower $O_3$ precursor

369 concentrations (Kleinman et al., 2002; Zaveri et al., 2003). The contribution of oceanic sources decreases

370 to 4% due to reduced emissions from shipping. The contribution from stratospheric $O_3$ (STR) is highest

371 in March (26%, 14 ppbv), lowest in August (7%, 3 ppbv) under present day conditions. Under net zero,

372 the highest contribution is increased to 39% (17 ppbv), and the lowest contribution is also increased, to

373 12% (3 ppbv). This may due to enhanced stratospheric circulation, slower photochemical loss and a

374 longer lifetime of $O_3$ in the troposphere allowing greater transport of stratospheric $O_3$ to the ground.

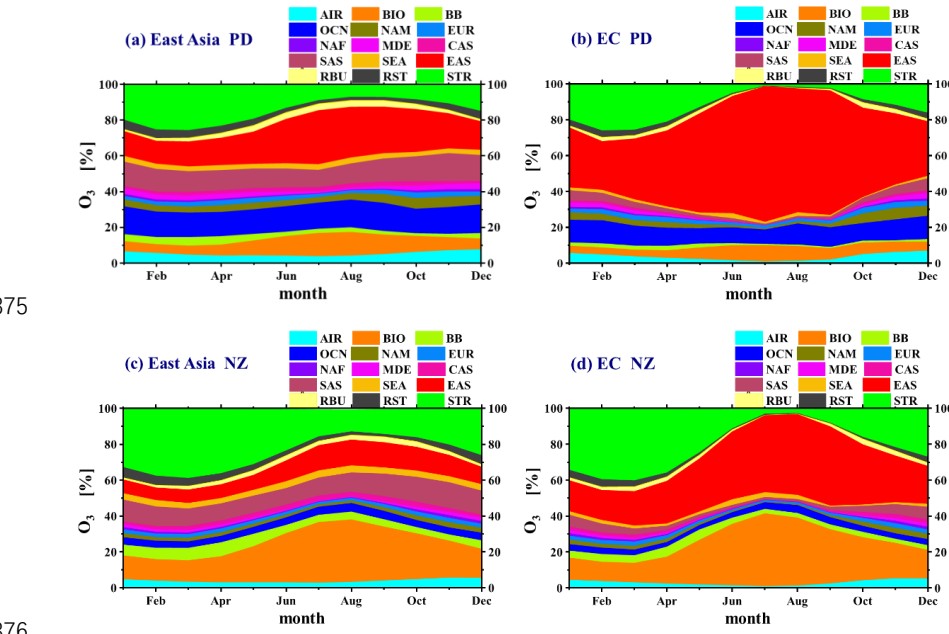

377 Figure 7 Contributions of different sources to surface $O_3$ under present day and net zero conditions over

378 East Asia (a, c) and Eastern China (b, d).

379 Over Eastern China (Figure 7b), the contribution from East Asian sources is highest, especially in

380 summer when it exceeds 70%. Biogenic, oceanic and South Asian sources make a smaller contribution

381 over this region, only 6%, 10%, and 3% on average, respectively. Under net zero (Figure 7d), the

382 contribution of East Asian sources drops to 42% in summer, but remains the dominant source of surface

383 $O_3$ in Eastern China. The contribution of biogenic sources is enhanced, especially in summer, reaching

384 40%, close to the contribution from East Asian sources. The stratospheric contribution (STR) is highest

385 in early spring (25%, 11 ppbv), and lowest in summer (2%, 1 ppbv). Under net zero, STR is enhanced to



40% (17 ppbv) in March and 3% (1 ppbv) in summer, similar to the seasonal contributions over East
Asia. In addition, the excess NO concentration in heavily urbanized Eastern China has a titration effect
on $O_3$, but the strong future decreases in NO weaken this effect, reducing the loss of stratospheric $O_3$ as
well as $O_3$ from local sources. Overall, surface $O_3$ shows substantial decreases through much of the year,
and the local contribution is reduced, which highlights the beneficial role that net zero policies may have
for controlling surface $O_3$ pollution in China.
**6 Summary and conclusions**
We quantify tropospheric $O_3$ budgets, spatiotemporal distributions of future surface $O_3$ in East Asia and
regional $O_3$ source contributions for 2060 under a net zero scenario, using the NCAR Community Earth
System Model (CESM) and online $O_3$ tagging methods. The simulated monthly mean global tropospheric
column $O_3$ and surface $O_3$ mixing ratios over East Asia capture the general features in observations well
under present day conditions. The offline simulations perform better than online simulations, as the
nudging provides a closer match to observed meteorological conditions. The tropospheric $O_3$ burden and
budget terms under present-day conditions in this study also matches those of previous model studies
well.
The simulated tropospheric $O_3$ burden is likely to decrease from 316 Tg under present day conditions to
247 Tg by 2060 under the net zero scenario. This brings it close to that found in previous studies under
preindustrial conditions of 239±22 Tg (Young et al., 2013). Future tropospheric $O_3$ chemical production
and loss are both reduced, and the net chemical tendency decreases from 397 to 81 $Tg(O_3)$ $yr^{-1}$. The
contribution of stratospheric $O_3$ increases from 69 to 77 Tg, due to enhancement of atmospheric
circulation and increased stratosphere-troposphere exchange caused by climate change and the longer
chemical lifetime of stratospheric $O_3$ in the troposphere under decreased anthropogenic emissions of
pollutants. The mean tropospheric lifetime of $O_3$ is increased by 2 days, ~10%. Over East Asia, one of
the highest anthropogenic emissions regions, the $O_3$ burden decreases from 25 to 19 Tg, and the net
chemical tendency drops from 227 to 137 $Tg(O_3)$ $yr^{-1}$. East Asia is a region of net $O_3$ production, and the
outflow is expected to decrease from 89 to 38 $Tg(O_3)$ $yr^{-1}$. The burden of $O_3$ from the stratosphere
increases from 5 to 6 Tg. The lifetime of tropospheric $O_3$ over East Asia is shorter than the global average,
~15 days, due to the high anthropogenic emissions, but increases by 2 days, similar to the global mean.
Compared with other SSP scenarios, particularly the much-studied SSP3-7.0 pathway, SSP1-1.9 provides
a more positive perspective on the opportunities for controlling future tropospheric $O_3$.
Regional average surface $O_3$ decreases throughout the year over East Asia, with highest decreases in
summer (16 ppbv) in the future under net zero scenario. Over Eastern China, the peak in surface $O_3$ shifts
from July to May. Surface $O_3$ decreases strongly in July (34 ppbv), and increases in winter, especially in
January, 12 ppbv. The increased $O_3$ in winter is caused by reduced titration of $O_3$ by NO associated with
lower anthropogenic NO emissions, and enhanced stratospheric input. The tropospheric $O_3$ over most
regions decreases due to the large decrease in $O_3$ precursors emissions. Climate change leads to only a
small increase in tropospheric $O_3$ under this scenario. Local anthropogenic emissions play a dominant
role in controlling $O_3$ changes over East Asia in summer, but this will drop substantially from 30% in



present day to 14% under net zero. The contribution of biogenic sources is enhanced, and forms the
dominant contributor to future surface $O_3$, especially in summer, ~40%. This enhanced contribution of
biogenic sources is due here to increased $O_3$ production efficiency associated with reduced $O_3$ precursors
concentrations, but may be underestimated if biogenic emissions also increase in future as expected. The
lower extent of climate change along SSP119 leads to relatively little impact on tropospheric $O_3$ under
net zero, while the emission reductions associated with net zero policies are sufficient to mitigate surface
$O_3$ pollution over East Asia, especially in summer.
The combined emissions and $O_3$ tagging method used here provide a reliable way to quantify the changes
of tropospheric $O_3$ and its sources in future under a net zero scenario. The results of this study clarify the
separate impacts of climate change and emissions on tropospheric $O_3$ changes over East Asia, and
highlight the significance of controlling $O_3$ precursors emissions along the net zero scenario, especially
anthropogenic emissions. The reduction of anthropogenic $O_3$ precursors emission should be the most
effective way to control the increase of tropospheric $O_3$, which requires joint efforts on a global scale.

**Competing interests**

The authors declare that they have no conflict of interest.

**Acknowledgements**

This study was supported by the National Key Research and Development Program of China (Grant No.,
2022YFC3701204) and the National Natural Science Foundation of China (Grant No., 42021004).
Numerical calculations in this paper have been performed on the high-performance computing system at
the High Performance Computing Center, Nanjing University of Information Science and Technology.
The authors would like to thank Prof. Tim Butler and Dr. Aurelia Lupascu at the Institute for Advanced
Sustainability Studies (now the Research Institute for Sustainability) in Potsdam, Germany for helping
us to update the TOAST source attribution code in the CESM model.

**Data availability**

CAQRA can be freely downloaded at https://doi.org/10.11922/sciencedb.00053, and the prototype
product, which contains the monthly and annual means of the CAQRA dataset, is available at
https://doi.org/10.11922/sciencedb.00092. The simulated $O_3$ data generated in this study are available
upon request.

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
