# Peer review of "Future tropospheric ozone budget and distribution over"

_EGUsphere, 2023_

## Referee Comment (RC2)

**Reviewer comments on Hou et al., "Future tropospheric ozone budget and distribution over East Asia under a Net Zero scenario"**

This study presents an assessment of the impact of net zero emission policies on tropospheric and surface ozone concentrations over east Asia using the Community Earth System Model (CESM). Changes in ozone are simulated between 2015 and 2060 using the SSP119, with emissions over eastern China from the Ambitious pollution-Neutral-goals scenario from the Dynamic Projection model. A tagging method is used to analyse the impact on ozone concentrations from different regional and sectoral source of NOx. Net zero policies are shown to reduce both the tropospheric ozone burden and surface ozone concentrations in the future, with small increases due to stratospheric exchange and climate change. Local NOx emission sources show a large reduction in the contribution to surface ozone concentrations in the future with contribution from biogenic sources increasing. The study shows the importance of future emission reductions to decreasing ozone concentrations and the importance of net-zero policies.

I found this paper well written with some interesting results, particularly the tagging, that cover an important area of policy. I think the manuscript is suitable for publication once the following comments have been addressed.

**Major Comments**

- At various points in the manuscript (introduction and results) more reference could be made to more recent CMIP6 studies e.g. Zanis et al., (2022) and Allen et al., (2020). See minor comments for more details.
- I think some more comments about the influence of climate change on the results would be useful. Firstly, there is no mention of what climate change signal is simulated by this model in the future e.g. what is the future temperature change globally and regionally over East Asia by 2060 in this future scenario? This would allow some context to be placed on the climate change signal in relation to the results of other studies e.g. Zanis et al., (2022). Also, it would be good for the author to comment on whether a simulation period of 15 years is long enough to be able to simulate a climate change signal in comparison to internal variability, particularly in a future pathway with a smaller assumed signal. Given that this is a single model study there is also likely to be larger differences between models in the simulation of the future climate change signal. Furthermore, is there an impact from using the different model configurations when calculating the impact of climate change on the results? A lot of mention is made of climate change influencing STE and ozone concentrations. Is this the only important mechanism of climate change impacts on ozone, what about temperature effects on chemical reaction rates?
- I think when talking about the results of the tagged ozone simulations it would be good to emphasise that the results are only due to changes in NOx and not other ozone precursors such as CO, VOC and CH4. In the results section of the manuscript, it could be interpretated that these future changes in NOx from different sources are the only influence on future ozone concentrations. It would be useful in the manuscript if more comment is made about the impact of changes in CO, VOCs and CH4 on ozone concentrations in future.
- The tropospheric ozone evaluation section could be expanded to include some of the reasons that the model might not reproduce the observations of ozone in both the troposphere and at the surface. Also is there a reason, apart from better meteorology, that the offline simulation is better able to reproduce observations than the online simulations (line 250)?

Furthermore, can the authors comment on the impact that model biases could have on the projection of future ozone concentrations in this study (see Liu et al., (2022b) for methods to correct biases)?

- The results presented here are from using a single model to simulate future changes in ozone concentrations. Given the range of projections made in previous multi-model assessment could more comment be made in the discussion on the impact of only using output from a single model on the simulated changes in future ozone concentrations?
- It would be good to provide more comment and discussion on whether the scale of the emission reductions are considered feasible in the SSP119 pathway, especially if they are shown to return tropospheric ozone burdens back down to pre-industrial levels? As part of this it would be good to place these results in more context with those of the more studied pessimistic pathway of SSP370, particularly the influence of climate change.

**Minor Comments**

Line 24 – need to highlight here (and other places) that the tagging results are only for NOx sources

Line 36 – could mention that these effects are important at the surface

Line 40 – shorter timescales than long-lived greenhouse gases

Line 57 – not just dependent on climate mitigation measures but air pollution policies

Line 66 – development storylines or socio-economic pathways? Also mention anthropogenic radiative forcing

Line 76 – what scenario are these increases for?

Line 91-92 – Can you say what has led to the large differences between the results of these studies?

Line 147 – You are using the 2015 ssp119 as present day emissions for 2015? How representative are these compared to other inventories for 2015 e.g. MEIC, especially when using the DPEC for future scenarios over China.

Line 159-160 – Related to the point above. How different are the DPEC emissions over China compared to SSP119 and why are they different? What is the added advantage of using this regional emission inventory compared to the global one?

Line 179 – Is it worth showing the aerosol precursor emission changes (SO2, BC, OC) in Table 2 if they do not directly impact ozone concentrations and the main message of the paper?

Line 186 – why only tagging NOx and not VOCs?

Line 225 – Could figure 2 also include a different plot to highlight the overestimation?

Line 265-267 – need to look at CMIP6 reference of Zanis et al., (2022)

Line 269-271 – why is the decrease in photochemical production less over East Asia than global?

Line 271-272 – can you define what you mean here by net outflow region? Is this ozone produced over East Asia is more favourably exported to other regions?

Line 278 – In Table 4 you could also include up to date numbers from the CMIP6 study by from Griffiths et al., (2021) https://doi.org/10.5194/acp-21-4187-2021

Line 306-309 – It would be good to show this comparison of emission inventories for both present day and future in section 2

Line 311-313 – It would be good to make more comparisons to Zanis et al., (2022) which shows a consistent increase over East Asia in these multi-model responses whereas here the difference shows a reduction in JJA over Eastern China due to climate change. Can you explain the differences here or is it due to a smaller climate change signal? Also see major comment on this.

Line 313-314 – Linked to the above. Is this all due to the STE increases? What about other effects e.g. temperature on surface O3 response. This could be expanded to include more discussion on other impacts.

Line 327-328 - is this changes in the tropopause height significant as hard to see on the figure?

Line 338-341 – Could this be expanded to say the decrease in anthropogenic sources shifts the seasonal cycle from summer towards spring, which is more dominated by STE?

Line 349-352 – Reference could be made Liu et al., (2022a) showing similar seasonal effects in another model (https://doi.org/10.5194/acp-22-1209-2022) and also a preprint by the same author studying Net Zero policies over China (https://doi.org/10.5194/egusphere-2023-230).

Line 355 – Figure 6 needs to be clearer here on what the bottom panel of figure c and d is and why is this included next to NOx concentrations? Could make NOx separate panels?

Line 361 – Could more be explained on what the dominant biogenic sources of NOx are? Also would biogenic sources of VOCs not be more important in future conditions for ozone formation?

Line 365 - Is the biogenic source enhanced of has the relative contribution (as anthropogenic reduced) increased?

Line 367 – Linked to the above point. Do the biogenic sources change between the present day and future scenarios?

Line 371 – Could the percentage change be linked to the actual ppbv change in the rest of this section like it is done here?

Line 359-374 – Could a comparison also be made of these to local vs external contributions to those in Fig 5 of Turnock et al., (2019)?

Line 377 – Can more detail be included in the caption of Figure 7 to say what simulations these results have been derived from and also to make clear that these changes are only due to NOx.

Line 389-391 – Is there a substantial change in the external sources to China in the future SSP119 pathway and does this influence ozone concentrations in East Asia?

Line 392 – Could more be made in the conclusion section to try and link the results to the impact on air quality and health?

Line 426-427 – Need to make sure this is clear that it is biogenic NOx sources considered here and also that they are not changing in this study.

Line 427-430 – I found this to be quite a broad statement that net zero policies are sufficient to mitigate surface ozone pollution over East Asia, especially in summer. Does this mean that there won't be any issues from ozone in summertime in the future under this scenario?

**References**

Allen, R. J., Turnock, S., Nabat, P., Neubauer, D., Lohmann, U., Olivié, D., et al. (2020). Climate and air quality impacts due to mitigation of non-methane near-term climate forcers. Atmospheric Chemistry and Physics, 20(16), 9641–9663. https://doi.org/10.5194/acp-20-9641-2020

Griffiths, P. T., Murray, L. T., Zeng, G., Shin, Y. M., Abraham, N. L., Archibald, A. T., Deushi, M., Emmons, L. K., Galbally, I. E., Hassler, B., Horowitz, L. W., Keeble, J., Liu, J., Moeini, O., Naik, V., O'Connor, F. M., Oshima, N., Tarasick, D., Tilmes, S., Turnock, S. T., Wild, O., Young, P. J., and Zanis, P.: Tropospheric ozone in CMIP6 simulations, Atmos. Chem. Phys., 21, 4187–4218, https://doi.org/10.5194/acp-21-4187-2021, 2021.

Liu, Z., Doherty, R. M., Wild, O., O'Connor, F. M., and Turnock, S. T.: Correcting ozone biases in a global chemistry–climate model: implications for future ozone, Atmos. Chem. Phys., 22, 12543–12557, https://doi.org/10.5194/acp-22-12543-2022, 2022a.

Liu, Z., Doherty, R. M., Wild, O., O'Connor, F. M., and Turnock, S. T.: Tropospheric ozone changes and ozone sensitivity from the present day to the future under shared socio-economic pathways, Atmos. Chem. Phys., 22, 1209–1227, https://doi.org/10.5194/acp-22-1209-2022, 2022b.

Turnock, S. T., Wild, O., Sellara, A., O'Connor, F. M.: 300 years of tropospheric ozone changes using CMIP6 scenarios with a parameterised approach. Atmos. Environ., 213, 686-698, https://doi.org/10.1016/j.atmosenv.2019.07.001, 2019.

Zanis, P., Akritidis, D., Turnock, S., Naik, V., Szopa, S., Georgoulias, A. K., et al. (2022). Climate change penalty and benefit on surface ozone: A global perspective based on CMIP6 Earth system models. Environmental Research Letters, 17, 024014. https://doi.org/10.1088/1748-9326/ac4a34

---

## Author Comment (AC1)

This study analyzed changes in the global tropospheric ozone budget, as well as the spatial-temporal variations of surface $O_3$ in East Asia. It investigated contributions from regional emissions, intercontinental transport, and climate changes in a scenario where emissions of $O_3$ precursors are reduced under future net-zero emissions policies based on CESM. The findings reveal that emission reductions led to a decrease in tropospheric $O_3$, while climate changes had only a minor impact. Additionally, the contribution of biological sources to surface ozone showed a gradual increase. The research underscores the significant co-benefits of net-zero policies targeting climate change in addressing surface $O_3$ pollution over East Asia. I recommend that this article be published with some minor modifications.

Response: We appreciate your time and effort for evaluating our work. We have made corresponding changes and revision in the updated version of the manuscript. To address the reviewer's comments in an organized manner, we have numbered the questions, and our responses are highlighted in blue. The reviewer's comments are presented in black. When referring to the manuscript, it is italicized.

Specific Comments:

1. Line 109 and Line 122: CESM has be clarified in Line 99, it can be abbreviated directly.
Response: Thanks for your suggestion. CESM are abbreviated directly in Line 110 and Line 123 in the updated manuscript.

2. Line 137: Should be "Modern Era Retrospective analysis for Research and Applications, version 2 (MERRA2)"
Response: Yes, we missed "version 2". It is added. Please refer to Line 137 in the updated manuscript.

3. Line 132-133: Why didn't you use CESM1 for both online and offline simulation.
Response: CESM2 contains more chemical species and chemical reactions, and stratospheric chemistry is included, which can be used to illustrate the changes in the contribution of emissions to stratospheric chemistry and stratospheric-tropospheric-transport. CESM2 is superior to CESM1 and is a better choice for research. However, tagging methods are only introduced to CESM1. Therefore, we have to use CESM1 for online simulation. In addition, we compared the offline simulation results of CESM2 and CESM1 in present day. There was no significant difference in the distribution of tropospheric ozone (Figure R1), especially for surface $O_3$. So we used offline experiments of CESM2. We gave the explanations in the text. Please refer to Line 129-133.

[Figure]

Figure R1 Annual tropospheric column $O_3$ (DU) from OMI/MLS (left), offline CESM1.2.2 (middle) and offline CESM2.2.0 (right) simulations under present day conditions.

4. Section 2.3: Ensure consistent tenses throughout Section 2.

Response: Thanks for your comment. We checked and revised the tenses in the whole updated manuscript.

5. Line 267-268: This explanation is confusing. A Longer chemical lifetime of stratospheric $O_3$ may reduce its contribution to tropospheric ozone burden.

Response: This is an ideal assumption. If the amount of stratospheric $O_3$ entering the troposphere and the amount of tropospheric ozone do not change, the longer the chemical life of stratospheric ozone entering the troposphere, the higher the cumulative amount of stratospheric ozone will be. This is the result that only considering the changes in the lifetime of stratospheric ozone after it enters the troposphere.

6. Line 272-273: How can weakened outflow reflect reduced regional production.

Response: Yes, it is improper to say weakened outflow reflect reduced regional production. This sentence is rewritten in the text (Line 271-273).

"The negative 'Residual' budget term for East Asia indicates that East Asia is a net outflow region for tropospheric $O_3$, and this outflow is weakened in the future, from 89 Tg($O_3$) yr$^{-1}$ under present day conditions to 38 Tg($O_3$) yr$^{-1}$ under net zero."

7. Line 311: Another similar table to Table 2, showing different results in winter and summer could support your attribution here.

Response: Thanks for your suggestion. It is a comparison between present day and net-zero. The decrease of NOx in month can be seen from Figure 6. We rewrite this sentence in the text (Line 311-312).

"This increase in surface $O_3$ is caused by a weakening of titration under lower regional NO emissions in the future."

8. Line 326-328: Can you explain why temperature increase and circulation enhance in the future? I think temperature should decrease in a net zero scenario.

Response: According IPCC AR6 and the interpretation of Zhou (2021), the temperature in 2060 still is increased compared with present day, through it will decrease after 2040 under SSP1-1.9 scenario. The increase of air temperature will enhance the movement of atmosphere. In the future, the circulation may be enhanced, which has been mentioned by Sudo et al. (2003).

[Figure]

Figure R2 Global surface air temperature changes from CMIP6 historical and scenario simulations (Source: IPCC AR6).

Zhou, T. J., Chen, Z. M., Chen, X. L. et al.: Interpreting IPCC AR6: future global climate based on projection under scenarios and on near-term information, Climate Change Research, 17(6): 652-663 https://doi.org/10.12006/j.issn.1673-1719.2021.239, 2021.

Sudo, K., Takahashi, M., and Akimoto, H.: Future changes in stratosphere-troposphere exchange and their impacts on future tropospheric ozone simulations, Geophys. Res. Lett., 30(24), 2256, https://doi.org/10.1029/2003GL018526, 2003

9. Figure 6(c) and (d) shows the NOx and $O_3$ chemical tendency, the explanation of the ordinate 'rate' needs to be clarified in more detail.

Response: We added an explanation about the relationship between $O_3$ chemical tendency and "Rate" in the updated manuscript (Line 358-359).

"*The net $O_3$ chemical tendency is net photochemical rate of $O_3$ change (kg/s).*"

10. Line 373-374: Are there any evidences from your simulations that can support this explanation?

Response: We have some evidence. As shown in Table 4, the photochemical loss of tropospheric $O_3$ is slower and the lifetime is longer in the future. The enhanced stratospheric circulation can be seen from Figure 5. These can support our explanation in this study. For more evidence, more detailed experiments should be done in the future.

---

## Author Comment (AC2)

**Reviewer comments on Hou et al., "Future tropospheric ozone budget and distribution over East Asia under a Net Zero scenario"**

This study presents an assessment of the impact of net zero emission policies on tropospheric and surface ozone concentrations over east Asia using the Community Earth System Model (CESM). Changes in ozone are simulated between 2015 and 2060 using the SSP119, with emissions over eastern China from the Ambitious pollution-Neutral-goals scenario from the Dynamic Projection model. A tagging method is used to analyse the impact on ozone concentrations from different regional and sectoral source of NOx. Net zero policies are shown to reduce both the tropospheric ozone burden and surface ozone concentrations in the future, with small increases due to stratospheric exchange and climate change. Local NOx emission sources show a large reduction in the contribution to surface ozone concentrations in the future with contribution from biogenic sources increasing. The study shows the importance of future emission reductions to decreasing ozone concentrations and the importance of net-zero policies. I found this paper well written with some interesting results, particularly the tagging, that cover an important area of policy. I think the manuscript is suitable for publication once the following comments have been addressed.

Response: We appreciate your time and effort for evaluating our work. We have made corresponding changes and revision in the updated version of the manuscript. The revision in updated manuscript is highlighted in yellow. To address the reviewer's comments in an organized manner, we have numbered the questions, and our responses are highlighted in blue. The reviewer's comments are presented in black. When referring to the manuscript, it is italicized.

**Major Comments**

1. At various points in the manuscript (introduction and results) more reference could be made to more recent CMIP6 studies e.g. Zanis et al., (2022) and Allen et al., (2020). See minor comments for more details.

Response: Thanks for your suggestions. We have added relevant previous research and cited the references according to the reviewer's suggestions. Please refer to Line 74-78 in the revised manuscript. More revisions in detail are listed in the response to the reviewer's minor comments.

2. I think some more comments about the influence of climate change on the results would be useful. Firstly, there is no mention of what climate change signal is simulated by this model in the future e.g. what is the future temperature change globally and regionally over East Asia by 2060 in this future scenario? This would allow some context to be placed on the climate change signal in relation to the results of other studies e.g. Zanis et al., (2022).

Also, it would be good for the author to comment on whether a simulation period of 15 years is long enough to be able to simulate a climate change signal in comparison to internal variability, particularly in a future pathway with a smaller assumed signal. Given that this is a single model study there is also likely to be larger differences between models in the simulation of the future climate change signal.

Furthermore, is there an impact from using the different model configurations when calculating the impact of climate change on the results? A lot of mention is made of climate change influencing STE and ozone concentrations. Is this the only important mechanism of climate

change impacts on ozone, what about temperature effects on chemical reaction rates?

Response: The climate change in this study involves a combination of changes in atmospheric parameters (air temperature, relative humidity, atmospheric circulation etc.), from the free-run of atmospheric simulation experiments (online). The climate change along SSP119 is much weaker than other pathways, and the change in global surface air temperature in this study is not significant (Figure R1). Over East Asia, the surface air temperature is increased by an annual average of 0.2°C. We have added the introduction in the updated manuscript (Line 327 to 332) and Supplementary Material (Figure S4). More refined experiments could be performed to accurately quantify the impact of each meteorological parameter change on ozone, but this attribution has not been explored in the current work as the overall effects are small.

[Figure]

Figure R1 Distribution of annual surface air temperature (K) in present day (left, PD) and net zero (middle, NZ), and their difference (right, NZ-PD). The values in the right corner of each sub-figure are the average on the globe.

Simulations of 30 years are preferred for climate studies, but we capture most of the variability within 15 years. The spin-up of more than ten years is enough for adjusting the balance of oxidation chemistry to the fixed $CH_4$ concentrations used here along the SSP1-1.9 pathway, and a similar approach was used in the studies of Liu et al. (2022b, 2023). The simulation period of 15 years is a compromise, given computational constraints.

Differences between models or different configurations of the same model both affect the simulation results. CESM showed excellent performance in CMIP6, and MOZART, the chemical module in CESM, is also a credible and well-tested parameterization scheme for atmospheric chemistry. Recent applications of CAM-chem have demonstrated its ability to represent climate changes (Eyring et al., 2016; Fan et al., 2020; Yang et al., 2021). And the use of similar chemical mechanism (MOZART) in different model versions may reduce the uncertainties in the simulation results (Line 118-119 and 139-140).

In this study, due to the changes of atmospheric circulation and temperature, the contribution of stratospheric $O_3$ to the troposphere increases, and $O_3S$ on surface also increases in the future. However, $O_3S$ on surface decreases under the impact of emission changes. The mechanisms governing how climate change affects surface ozone are complex. The change of temperature not only has a direct effect on the chemical production of ozone, but also affects atmospheric circulation, affecting vertical and horizontal transport processes, and on relatively humidity which strongly influences the chemical lifetime of ozone. In the updated manuscript, we have added further explanation of these points (Line 283-284, 327-332, 346-347).

Eyring, V., Bony, S., Meehl, G. A., Senior, C. A., Stevens, B., Stouffer, R. J., and Taylor, K. E.: Overview of the Coupled Model Intercomparison Project Phase 6 (CMIP6) experimental design and organization, Geosci. Model Dev., 9, 1937–1958, https://doi.org/10.5194/gmd-9-1937-2016, 2016.

Fan, X., Duan, Q., Shen, C., Wu, Y., and Xing, C.: Global surface air temperatures in CMIP6: historical performance and future changes, Environ. Res. Lett., 15, 104056, https://doi.org/10.1088/1748-9326/abb051, 2020.

Yang, X. L., Zhou, B. T., Xu, Y., and Han, Z.-Y.: CMIP6 Evaluation and Projection of Temperature and Precipitation over China, Adv. Atmos. Sci., 38(5), 817−830, https://doi.org/10.1007/s00376-021-0351-4, 2021.

3. I think when talking about the results of the tagged ozone simulations it would be good to emphasise that the results are only due to changes in NOx and not other ozone precursors such as CO, VOC and CH4. In the results section of the manuscript, it could be interpretated that these future changes in NOx from different sources are the only influence on future ozone concentrations. It would be useful in the manuscript if more comment is made about the impact of changes in CO, VOCs and CH4 on ozone concentrations in future.

Response: Thanks for your suggestions. We have now emphasized in the text that this tagging method only tracks the contribution of changes in NOx emissions on ozone (Line 203-205), although we note that changes in other precursors are already included in the chemistry, and so affect the tagged ozone, even though their contributions aren't explicitly tagged. The revisions in detail are listed in the response to the reviewer's minor comments below.

4.The tropospheric ozone evaluation section could be expanded to include some of the reasons that the model might not reproduce the observations of ozone in both the troposphere and at the surface. Also is there a reason, apart from better meteorology, that the offline simulation is better able to reproduce observations than the online simulations (line 250)? Furthermore, can the authors comment on the impact that model biases could have on the projection of future ozone concentrations in this study (see Liu et al., (2022b) for methods to correct biases)?

Response: The model biases may come from accuracy of the emissions inventory, resolution of simulation experiments, parameters in the model and other uncertainties in model and simulation experiments. Any one of these reasons may impact the projection of future ozone concentrations. The methods to correct biases could give us good results for the present day, but it is hard to know the performance under future conditions. The results of Liu et al. (2022b) show that the ozone sensitivity to changing emissions and climate may be overestimated with global chemistry–climate models. According to this result, the changes of tropospheric ozone in offline simulation may be overestimated in our results. However, this overestimation may be offset by the underestimation behavior of the offline simulation which is shown in Figure 2.

We have added some additional text on model biases. The accuracy of the emissions inventory we used in this study may affect the simulation results, especially at the surface. We have added an explanation in the tropospheric ozone evaluation section (Line 241-244). In the offline simulation, there are 56 levels in the vertical, while there are only 26 levels in the online simulation. The coarser vertical resolution will affect the calculation of vertical transport of atmospheric constituents, and may lead to biases in the simulation of tropospheric ozone in the

online simulation (Lines 265-266).

5.The results presented here are from using a single model to simulate future changes in ozone concentrations. Given the range of projections made in previous multi-model assessment could more comment be made in the discussion on the impact of only using output from a single model on the simulated changes in future ozone concentrations?

Response: Differences in models will lead to diverse simulation results. Multi-model assessment could show us the performance of each model or certain simulation settings. CESM showed excellent performance in CMIP6, and MOZART, the chemical module in CESM, is also a credible and well-tested parameterization scheme for atmospheric chemistry. Griffiths et al. (2021) shows that the changes of tropospheric-ozone budget terms in CESM match the results in GFDL-ESM4, MRI-ESM2-0, UKESM1 well, e.g., burden increase of 54 Tg[$O_3$] along SSP370 over 2000-2050 compared with average of 49 Tg[$O_3$] over the other four models. Therefore, our results should be credible.

6.It would be good to provide more comment and discussion on whether the scale of the emission reductions are considered feasible in the SSP119 pathway, especially if they are shown to return tropospheric ozone burdens back down to pre-industrial levels? As part of this it would be good to place these results in more context with those of the more studied pessimistic pathway of SSP370, particularly the influence of climate change.

Response: We have added comparisons with pre-industrial levels in Table 4 and additional description about the comparisons with pre-industrial levels and SSP3-7.0 has now also been included in the updated manuscript. Please refer to line 285 to 290.

**Minor Comments**

Line 24 – need to highlight here (and other places) that the tagging results are only for NOx sources

Response: We revised this sentence in the abstract according to the reviewer's comment (Line 26).

"*The contribution of biogenic NO sources is enhanced, and forms the dominant contributor to future surface $O_3$, especially in summer, ~40%.* "

Line 36 – could mention that these effects are important at the surface

Response: We modified this sentence in the updated manuscript (Line 37-40).

"*This excess $O_3$ acts as a pollutant and greenhouse gas, contributing to harmful smog that damages human health and ecosystems (Jerrett et al., 2009; Malley et al., 2017; Emberson, 2020) and contributing to higher tropospheric temperatures (Myhre et al., 2013; Stevenson et al., 2013).*"

Line 40 – shorter timescales than long-lived greenhouse gases

Response: We added "than the long-lived greenhouse gases such as $CO_2$" to the end of this sentence (Line 42-43).

Line 57 – not just dependent on climate mitigation measures but air pollution policies

Response: We modified the sentence according to the reviewer's comment (Line 59-60). "*Whether tropospheric $O_3$ increases or decreases in future is dependent on the climate mitigation measures and air pollution policies that are implemented.*"

Line 66 – development storylines or socio-economic pathways? Also mention anthropogenic radiative forcing

Response: We modified the sentence according to the reviewer's comment (Line 67-70). "*The five most widely-used scenarios are SSP1-1.9, SSP1-2.6, SSP2-4.5, SSP3-7.0, and SSP5-8.5, where SSP1-SSP5 represent differing socio-economic pathways and the suffix 1.9~8.5 indicates the total radiative forcing (W/m$^2$) at the end of the 21st century compared with that before the Industrial Revolution.*"

Line 76 – what scenario are these increases for?

Response: They increased for IPCC A1B (Wang et al., 2013), RCP6.0 (Zhu and Liao, 2016) and RCP4.5 (Hong et al., 2019) scenarios. We have improved the description (Line 83-84). "*In East Asia, surface $O_3$ has increased rapidly since 2000 (Lu et al., 2020), and is expected to increase by another ~10 ppbv by 2050 following the IPCC A1B (Wang et al., 2013), RCP6.0 (Zhu and Liao, 2016) and RCP4.5 (Hong et al., 2019) scenarios.*"

Line 91-92 – Can you say what has led to the large differences between the results of these studies?

Response: Many reasons may lead to the differences between studies, for example, the emission and climate scenarios that were used. Xu et al. (2022) used DPEC Ambitious-pollution-neutral-goals emission and RCP2.6 climate scenario, while Wang and Liao (2022) used SSP1-1.9 scenario. In addition, the models used are also different. Xu et al. (2022) used RegCM-Chem-YIBs, while Wang and Liao (2022) used the GEOS-Chem model.

We have revised the description in the updated manuscript. Please refer to line 94 to 99.

Line 147 – You are using the 2015 ssp119 as present day emissions for 2015? How representative are these compared to other inventories for 2015 e.g. MEIC, especially when using the DPEC for future scenarios over China.

Response: We have compared emissions on the SSP1-1.9, CMIP6 historical, and DPEC Ambitious-pollution-Neutral-goals scenarios, and plotted Figure R2 based on the data from International Institute for Applied Systems Analysis (IIASA, https://tntcat.iiasa.ac.at/SspDb/dsd?Action=htmlpage&page=50), below. This reveals that the SSP emissions in 2015 are essentially the same as the historical ones in CMIP6, and are thus suitable for the present day.

The DPEC anthropogenic emissions are based on SSP scenarios and MEIC, but give more anthropogenic sources of emissions and higher resolution in China (Tong et al., 2020; Cheng et al., 2021). We compared the anthropogenic emissions of NOx and VOCs on DPEC Ambitious-pollution-Neutral-goals scenario and MEIC in 2015. The differences in NOx and VOCs is 0.4 Tg and 1.8 Tg, respectively. There is little difference between DPEC Ambitious-pollution-Neutral-goals scenario and MEIC in present day emission.

In addition, we compared the anthropogenic emissions of some species (NO, CO, $C_2H_4$) on

SSP1-1.9 and DPEC Ambitious-pollution-Neutral-goals scenario in China, shown in Figure R3. The total anthropogenic NO emissions in DPEC Ambitious-pollution-Neutral-goals scenario are 1.1 Tg lower than in SSP1-1.9 in 2015, but higher over most regions of Eastern China. Please refer to line 170 to 174 in the updated manuscript.

[Figure]

Figure R2 The emissions of $CO_2$, $CH_4$, VOCs and NOx under SSPs scenarios from 2015 to 2100.

[Figure]

[Figure]

Figure R3 The comparisons of annual emissions inventories between SSP1-1.9 (left) and DPEC Ambitious-pollution-Neutral-goals scenario (middle) in 2015 (PD) and their differences (right, DPEC-SSP119) over China.

Cheng, J., Tong, D., Zhang, Q., Liu, Y., Lei, Y., Yan, G., Yan, L., Yu, S., Cui, R. Y., Clarke, L., Geng, G, N., Zheng, B., Zhang, X, Y., Davis, J, S., and He, K, B.: Pathways of China's PM2.5 air quality 2015–2060 in the context of carbon neutrality, Natl. Sci. Rev., nwab078, https://doi.org/10.1093/nsr/nwab078, 2021.

Tong, D., Cheng, J., Liu, Y., Yu, S., Yan, L., Hong, C., Qin, Y., Zhao, H., Zheng, Y., Geng, G., Li, M., Liu, F., Zhang, Y., Zheng, B., Clarke, L., and Zhang, Q.: Dynamic projection of anthropogenic emissions in China: methodology and 2015–2050 emission pathways under a range of socio-economic, climate policy, and pollution control scenarios, Atmos. Chem. Phys., 20, 5729–5757, https://doi.org/10.5194/acp-20-5729-2020, 2020.

Line 159-160 – Related to the point above. How different are the DPEC emissions over China compared to SSP119 and why are they different? What is the added advantage of using this regional emission inventory compared to the global one?

Response: The DPEC anthropogenic emissions are based on SSP scenarios and MEIC, but give more anthropogenic sources and higher resolution in China which more accurately characterize China's emission sources and the recent rapid changes in China's emissions. Therefore, DPEC emissions should be more suitable for the research in China than SSP scenarios. Please refer to line 170 to 174 in the updated manuscript.

Line 179 – Is it worth showing the aerosol precursor emission changes (SO2, BC, OC) in Table 2 if they do not directly impact ozone concentrations and the main message of the paper?

Response: $SO_2$, BC and OC have less direct impact on $O_3$. We have therefore deleted these aerosol precursor emission changes in Table 2 of the updated manuscript as the reviewer suggests.

Line 186 – why only tagging NOx and not VOCs?

Response: Over most of the world, ozone production is NOx-limited, although it is sensitive to VOCs in urban areas (Liu et al., 2022b). So we assume that changes in ozone over large areas are mainly affected by changes in NOx, and therefore tag NOx. Of course, this method does not represent all production, especially over VOCs-limited areas (Butler et al., 2018; Butler etal., 2020). In future studies, we could extend the tracking method and simulation experiments to further clarify the impact of changes on ozone in the future.

Butler, T., Lupascu, A., Coates, J., and Zhu, S.: TOAST 1.0: Tropospheric Ozone Attribution of Sources with Tagging for CESM 1.2.2, Geosci. Model Dev., 11, 2825–2840, https://doi.org/10.5194/gmd-11-2825-2018, 2018.

Butler, T., Lupascu, A. and Nalam A.: Attribution of ground-level ozone to anthropogenic and natural sources of nitrogen oxides and reactive carbon in a global chemical transport model, Atmos. Chem. Phys., 20, 10707–10731, https://doi.org/10.5194/acp-20-10707-2020, 2020.

Line 225 – Could figure 2 also include a different plot to highlight the overestimation?

Response: We have replaced the figures with the bias plot. The regions with overestimated values are easier to locate on this figure. Please see Figure 2 in the updated manuscript.

Line 265-267 – need to look at CMIP6 reference of Zanis et al., (2022)

Response: Following this suggestion, we have included a mention of increased stratospheric ozone that was also seen in the Zanis et al. study (Line 283-284).

Line 269-271 – why is the decrease in photochemical production less over East Asia than global?

Response: The main reason may be the higher emissions and smaller reductions of precursors over East Asia than the global average. We have added the explanation about it in the updated manuscript (Line 293).

Line 271-272 – can you define what you mean here by net outflow region? Is this ozone produced over East Asia is more favorably exported to other regions?

Response: We have added the explanation about "net outflow" in the updated manuscript (Line 294-298).

*"The negative "Residual" budget term for East Asia indicates that the production is larger than the sink, and the total contribution of vertical and horizontal transport from outside of East Asia is negative. This indicates that there is net outflow from East Asia with transport of tropospheric $O_3$ to other regions, and this outflow is weakened in the future, from 89 $Tg(O_3)$ $yr^{-1}$ under present day conditions to 38 $Tg(O_3)$ $yr^{-1}$ under net zero."*

Line 278 – In Table 4 you could also include up to date numbers from the CMIP6 study by from Griffiths et al., (2021) https://doi.org/10.5194/acp-21-4187-2021

Response: We have now included the results of tropospheric $O_3$ from Griffith et al. (2021) in Table 4, and related analysis are added in the updated manuscript (Line 280, 285-290).

Line 306-309 – It would be good to show this comparison of emission inventories for both present day and future in section 2

Response: We have calculated the anthropogenic emission in Ambitious-pollution-Neutral-goals scenario from DPEC and SSP1-1.9 scenario in China in present day and 2060. The total anthropogenic emissions of most species in DPEC is lower than SSP1-1.9. Such as total anthropogenic NO emission, it is 1.1 Tg $yr^{-1}$ lower in present day and 1.5 Tg $yr^{-1}$ lower in 2060. We have added the evidence to the updated manuscript (Line 172-174).

Line 311-313 – It would be good to make more comparisons to Zanis et al., (2022) which shows

a consistent increase over East Asia in these multi-model responses whereas here the difference shows a reduction in JJA over Eastern China due to climate change. Can you explain the differences here or is it due to a smaller climate change signal? Also see major comment on this.

Response: The residual figures in Figure 4 are the differences between online and offline simulations, which include slight differences in model setup and the impact of climate change. The differences in model setup between online and offline simulation and the smaller climate change signal may be the key reasons for the reduction in JJA over Eastern China in the right panel of Figure 4. Under net zero scenario, the change of climate along SSP119 is much weaker than along the SSP370 pathway that Zanis et al., (2022) used. As you mentioned, the smaller climate change signal may lead to different changes in polluted regions.

Line 313-314 – Linked to the above. Is this all due to the STE increases? What about other effects e.g. temperature on surface O3 response. This could be expanded to include more discussion on other impacts.

Response: The increase of STE is one reason for the tropospheric $O_3$ changes. The air temperature changes also have an impact on the changes of tropospheric $O_3$. Over East Asia, the air temperature is increased slightly (Figure R1) which may enhance the efficiency of precursor emissions to generate surface ozone in polluted regions (Zanis et al., 2022).

Line 327-328 - is this changes in the tropopause height significant as hard to see on the figure?

Response: The changes in the tropopause height are not significant in our simulation result at low latitudes. But the changes are significant in the mid-latitude, 7 hPa higher, which may be substantially impact the changes in stratospheric contribution (Line 359-361).

Line 338-341 – Could this be expanded to say the decrease in anthropogenic sources shifts the seasonal cycle from summer towards spring, which is more dominated by STE?

Response: Thanks for the suggestion. We have added more description about this part in the updated manuscript (Line 375-376).

Line 349-352 – Reference could be made Liu et al., (2022a) showing similar seasonal effects in another model (https://doi.org/10.5194/acp-22-1209-2022) and also a preprint by the same author studying Net Zero policies over China (https://doi.org/10.5194/egusphere-2023-230).

Response: Thanks for the suggestion. We have cited these two references in the updated manuscript (Line 386).

Line 355 – Figure 6 needs to be clearer here on what the bottom panel of figure c and d is and why is this included next to NOx concentrations? Could make NOx separate panels?

Response: NOx concentrations have a tight connection with net $O_3$ chemical tendency, and so we included them in the same panels. In the updated manuscript, we have put NOx in separate panels (c) and (d), while net $O_3$ chemical tendency are now in (e) and (f).

Line 361 – Could more be explained on what the dominant biogenic sources of NOx are? Also would dominant biogenic sources of NOx not be more important in future conditions for ozone formation?

Response: In this study, we consider NO soil emissions as the dominant biogenic sources of NOx following Emmons et al. (2020). In present day, the contribution of NOx biogenic sources to $O_3$ is less than the contribution of anthropogenic sources, especially in summer. In the future, the contribution of NOx biogenic sources to $O_3$ will increase significantly, and play a more important role. We have clarified this in line 399-400 and 404-406 in the updated manuscript.

Emmons, L. K., Schwantes, R. H., Orlando, J. J., Tyndall, G., Kinnison, D., Lamarque, J.-F., et al. (2020). The Chemistry Mechanism in the Community Earth System Model version 2 (CESM2). *Journal of Advances in Modeling Earth Systems*, 12, e2019MS001882. https://doi.org/10.1029/2019MS001882

Line 365 - Is the biogenic source enhanced of has the relative contribution (as anthropogenic reduced) increased?
Response: The annual contributions of biogenic source in present day are 3.8 ppbv over East Asia and 2.9 ppbv in Eastern China. In net zero (2060), they are 6.2 ppbv over East Asia and 8.8 ppbv in Eastern China. The absolute contribution of biogenic source is increased. We have added the absolute values of contributions in the updated manuscript (Line 400 and 406).

Line 367 – Linked to the above point. Do the biogenic sources change between the present day and future scenarios?
Response: In this study, the biogenic sources of NO, CO and VOCs are fixed in our experiments, and this has been stated in Section 2.2 (Line 182-184). The sources don't change, and these are listed in Table 2.

Line 371 – Could the percentage change be linked to the actual ppbv change in the rest of this section like it is done here?
Response: Thanks for your suggestions. We have added the actual ppbv changes in the manuscript (highlighted values from line 396 to 410)

Line 359-374 – Could a comparison also be made of these to local vs external contributions to those in Fig 5 of Turnock et al., (2019)?
Response: We have added Figure 8 to the updated manuscript according to the reviewer's suggestion. In this study, we only compare the differences in present day (2015) and net zero (2060), so the distributions of the key source contributions on surface $O_3$ (ppbv) are plotted. Figure 8 shows additional values to support the analysis of Figure 7.

Line 377 – Can more detail be included in the caption of Figure 7 to say what simulations these results have been derived from and also to make clear that these changes are only due to NOx.
Response: We have revised the caption of Figure 7 in the updated manuscript.
"*Figure 7 Contributions of different sources to surface $O_3$ under present day and net zero conditions over East Asia (a, c) and Eastern China (b, d). Results are from the online simulations (online-PD and online-NZ). 10 geographical source regions are divided from anthropogenic NOx emission. BIO, BB, AIR, and LIG are the contribution of NOx emission from biogenic sources, biomass burning, aircraft and lightning to $O_3$. STR is the contribution*

*of O₃ originating in the stratosphere.* "

Line 389-391 – Is there a substantial change in the external sources to China in the future SSP119 pathway and does this influence ozone concentrations in East Asia?

Response: As shown in Figure 8, the total contributions of anthropogenic NO emissions outside East Asia shows little change, decreasing by 0.2 ppbv. In the west of East Asia, the impact is more significant. This is now shown in Figure 8.

Line 392 – Could more be made in the conclusion section to try and link the results to the impact on air quality and health?

Response: Thanks for the suggestion. We have made a simple link with the impact on air quality in the conclusions in the updated manuscript (Line 469-470).

Line 426-427 – Need to make sure this is clear that it is biogenic NOx sources considered here and also that they are not changing in this study.

Response: Thanks for the comment. We have checked the biogenic sources of both NOx and VOCs, and make sure they are not changing in this study.

Line 427-430 – I found this to be quite a broad statement that net zero policies are sufficient to mitigate surface ozone pollution over East Asia, especially in summer. Does this mean that there won't be any issues from ozone in summertime in the future under this scenario?

Response: According to our results, surface ozone pollution will be mitigated over East Asia, especially Eastern China in summer, but it is hard to say that there won't be any issues from ozone in summertime in the future under this scenario, particularly in heavily populated urban regions which we may not be able to resolve in this study. It would be very interesting to explore this in more detail with a high resolution model in future studies.

**References**

Allen, R. J., Turnock, S., Nabat, P., Neubauer, D., Lohmann, U., Olivié, D., et al. (2020). Climate and air quality impacts due to mitigation of non-methane near-term climate forcers. Atmospheric Chemistry and Physics, 20(16), 9641–9663, https://acp.copernicus.org/articles/20/9641/2020/.

Griffiths, P. T., Murray, L. T., Zeng, G., Shin, Y. M., Abraham, N. L., Archibald, A. T., Deushi, M., Emmons, L. K., Galbally, I. E., Hassler, B., Horowitz, L. W., Keeble, J., Liu, J., Moeini, O., Naik, V., O'Connor, F. M., Oshima, N., Tarasick, D., Tilmes, S., Turnock, S. T., Wild, O., Young, P. J., and Zanis, P.: Tropospheric ozone in CMIP6 simulations, Atmos. Chem. Phys., 21, 4187–4218, https://doi.org/10.5194/acp-21-4187-2021, 2021.

Liu, Z., Doherty, R. M., Wild, O., O'Connor, F. M., and Turnock, S. T.: Correcting ozone biases in a global chemistry–climate model: implications for future ozone, Atmos. Chem. Phys., 22, 12543–12557, https://doi.org/10.5194/acp-22-12543-2022, 2022a.

Liu, Z., Doherty, R. M., Wild, O., O'Connor, F. M., and Turnock, S. T.: Tropospheric ozone changes and ozone sensitivity from the present day to the future under shared socio-economic pathways, Atmos. Chem. Phys., 22, 1209–1227, https://doi.org/10.5194/acp-22-1209-2022, 2022b.

Turnock, S. T., Wild, O., Sellara, A., O'Connor, F. M.: 300 years of tropospheric ozone changes using CMIP6 scenarios with a parameterised approach. Atmos. Environ., 213, 686-698, https://doi.org/10.1016/j.atmosenv.2019.07.001, 2019.

Zanis, P., Akritidis, D., Turnock, S., Naik, V., Szopa, S., Georgoulias, A. K., et al. (2022). Climate change penalty and benefit on surface ozone: A global perspective based on CMIP6 Earth system models. Environmental Research Letters, 17, 024014, https://doi.org/10.1088/1748-9326/ac4a34.